# Development of Eco-Efficient Composite from Textile Waste with Polyamide Matrix

**DOI:** 10.3390/polym16142061

**Published:** 2024-07-19

**Authors:** Francisco Burgada, Marina P. Arrieta, Begoña Borrell, Octavio Fenollar

**Affiliations:** 1Textile Research Institute (AITEX), Carretera de Banyeres, 10, 03802 Alcoy, Spain; fburgada@aitex.es (F.B.); bborrell@aitex.es (B.B.); 2Grupo de Investigación Polímeros, Caracterización y Aplicaciones (POLCA), 28006 Madrid, Spain; m.arrieta@upm.es; 3Departamento de Ingeniería Química Industrial y del Medio Ambiente, Escuela Técnica Superior de Ingenieros Industriales, Universidad Politécnica de Madrid (ETSII-UPM), C/José Gutiérrez Abascal 2, 28006 Madrid, Spain; 4University Research Institute of Materials Technology (ITM), Universitat Politècnica de València (UPV), Plaza Ferrándiz y Carbonell 1, 03801 Alcoy, Spain

**Keywords:** textile waste, revalorization, ecoefficiency, composite, polyamide

## Abstract

The main aim of the present work is to evaluate and characterize the mechanical, morphological and thermal properties of wastes coming from the textile industry, mainly composed of cotton and polyester. These wastes will be thereafter implemented in commodity plastic such as polyamide, in order to develop new formulations of environmentally friendly materials. The composites were produced by extrusion and injection-molded processes in amounts between 15 wt.% and 60 wt.% of textile waste. With the objective of improving the properties of the materials, silanes were used as a compatibilizer between the textile fibers and the polymeric matrix. The effect of the compatibilizer in the composites was studied together with the effect of the amount of textile fiber added to the composites. Mechanical, thermal, morphological and wettability properties were analyzed for each composite. The results show that the use of silanes improves the interaction especially in those composites with a higher amount of textile waste, offering a balanced mechanical behavior with significantly high quantities. On the other hand, the melting temperature does not vary significantly with the introduction of silanes and textile waste content, although the incorporation of textile waste slightly reduces up to 23% the degradation temperature of the resulting composites. The wettability of the composites is also increased up to 16% with the incorporation of textile waste. Finally, the appearance of the composites with textile waste is strongly influenced by the incorporation of the reinforcement, offering shades close to dark brown in the whole range of compositions.

## 1. Introduction

The overwhelming consumption of clothing, which is likely to have been increasing in recent years, has a huge environmental impact and generates a large amount of waste, both during the manufacture of the products and the waste remaining after the useful life of the garments. On the other hand, the environmental impacts of textile and clothing consumption in the EU are difficult to estimate due to their diversity and the fact that they occur all over the world. A large amount of fiber is lost during the production of fibers and garments due to the shedding of natural fibers. Garment manufacturing, including cutting and sewing, also produces large amounts of fabric waste [1,2]. This fabric can be reused or recycled if factories take it into account and provided that suitable markets can be accessed. Existing routes to reuse and recycle the surplus leave plenty of scope for capturing more value and reducing waste from the production process, thus reducing demand for primary products [3,4]. It is in this context that the concept of waste revalorization becomes important as one of the measures to reduce the environmental impact of different human-generated wastes [5,6].

Taking into account that the textile industry uses mostly fibers such as cotton for the vast majority of production, and that in recent years the trend is towards an increase in the use of synthetic fibers such as polyester, it is possible to foresee that this trend of including more and more synthetic fibers in fabrics will increase, making the textile waste generated by the industry more and more interesting for addition as filler to different matrices of polymeric materials [7,8,9,10]. This will improve the current revaluation of this waste, which is mainly used for energy recovery. In many cases, recycled textile wastes, especially those that have undergone multiple recycling cycles, suffer a significant decrease in their properties, which makes it impossible to use them as the only material in a composition [11,12]. This problem has given rise to the development of a new form of waste revaluation through the creation of polymer matrix composites, where a polymer acts as a matrix and provides the main properties, and the waste serves as a filler to reduce the cost of the mixture without significantly worsening the overall properties of the composite [13,14]. It is remarkable that one of the new applications of fiber-reinforced composite materials from matrix waste is their use in the area of 3D printing [15,16].

In this case, such impact reduction is intended to be achieved by combining a waste from the textile industry as a material to be used in combination with a matrix composed of a thermoplastic polymer such as polyamide 6 [17,18], thus achieving the designation of eco-efficient composite. These materials have a high potential to address environmental issues, enhance material properties, offer economic benefits, and align with sustainability and innovation goals.

Although the chosen matrix has a nature similar to that of textile waste, this does not guarantee its miscibility, nor the good behavior of the final composite material. It is in this context that the need to evaluate the use of coupling agents arises. Coupling agents are additives that provide a stable chemical bond between different materials, generally between inorganic and organic products, thus improving the interaction between the matrix and the fibers [19,20]. This bond stability leads to significant improvements in the properties of the composite [21,22,23].

For this work, we have chosen to evaluate the addition of silanes as coupling agents, since they are the most common and industrially developed coupling agents. Silanes can present different hydrolyzable groups, called alkoxy groups, and these represent the reactivity of that silane, and therefore its ability to anchor to the surface of the inorganic component [24,25]. As for the way of incorporating these coupling agents, there are two ways of adding them to the composite material: (a) incorporation prior to the reinforcement substrate by means of an independent treatment, which results in a composite with better performance but requires a series of previous stages that make the manufacturing process more expensive [26,27]; (b) incorporation of the coupling agent as an additive to the polymeric matrix [28,29], thus simplifying the process at the cost of less than optimal properties but reducing the manufacturing cost compared to the previous method.

This work introduces an innovative approach to developing eco-efficient composite materials that not only promotes the reuse of textile waste, thereby reducing environ-mental impact, but also offers an economical and sustainable solution for the textile and polymer industries. In addition, composites were prepared through extrusion and injection molding, and their properties were characterized through mechanical, morphological, and thermal tests. The results showed good fiber dispersion, enhanced mechanical performance, and thermal stability, thus demonstrating a sustainable and economical solution for reusing textile waste in the polymer industry.

## 2. Materials and Methods

### 2.1. Materials

The polyamide (PA) used was Polyamide 6 with a grade of GP1100A (W), supplied in pellet form by Songhan Plastic Technology Co. (Shanghai, China) and manufactured by LG Chemical (Seoul, Republic of Korea). As the coupling agent used, Silane N-(n-butyl)-3-aminopropyltrimethoxysilane was supplied by Evonik Resource Efficiency GmbH (Hanau-Wolfgang, Germany) with the trade name of Dynasylan 1189. As the filler used, a textile waste from post-consumption of standard garments made with woven fabric (Figure 1) was supplied by Textile Industry Research Association AITEX (Alcoy, Spain). The composition of textile waste used is described in Table 1. The average fiber diameter of textile waste was 14 microns, and the fiber length ranged from 0.8 to 4 mm.

### 2.2. Sample Preparation

Polyamide and textile waste were dried separately at 60 °C for 48 h in a drying oven Carbolite Eurotherm 2416 CG (Hope Valley, UK) in order to remove any residual moisture prior to processing. Polyamide in pellet form and textile waste in the form of short fibers were mixed prior to being fed into the main hopper of a co-rotating twin-screw extruder (Construcciones Mecanicas Dupra, S.L., Alicante, Spain). This extruder machine has a 25 mm diameter with a length-to-diameter ratio (L/D) of 24. The extrusion process was carried out at a rotating speed of 20 rpm, setting the temperature profile, from the hopper to the die, as follows: 220–230–240–250 °C. The different composites were extruded through a round die to produce strands, which were pelletized using an air-knife unit. In all cases, residence time was approximately 1 min. Table 2 shows all compositions considered in this work.

The compounded pellets were, thereafter, shaped into standard samples by injection molding in a Meteor 270/75 from Mateu & Solé (Barcelona, Spain). The temperature profile in the injection molding unit was 235 °C (hopper), 240 °C, 245 °C, 250 °C (injection nozzle). A clamping force of 75 tons was applied, while the cavity filling and cooling times were set to 1 and 10 s, respectively. After obtaining the injected specimens, they were taken to the cutting machine to remove the remains of the injection ducts and burrs, obtaining the standardized specimens for the impact tests (80 × 10 × 4 mm^3^) and the standardized specimens for the tensile test (150 × 10 × 4 mm^3^).

### 2.3. Material Characterization

#### 2.3.1. Mechanical Tests

Tensile tests of injection-molded specimens were carried out in a universal testing machine ELIB 50 from S.A.E. Ibertest (Madrid, Spain) according to ISO 527-1:2012. A 5 kN load cell was used, and the cross-head applied was of 5 mm·min^−1^. Impact strength was determined using a Charpy pendulum from Metrotec SA (San Sebastián, Spain) with notched samples, applying an energy of 6 J, following the guidelines of ISO 179-1:2010. Shore D hardness was measured with a model 676-D durometer from J Bot Instruments (Barcelona, Spain) according to ISO 868:2003. All samples were tested under ambient conditions (23 °C/50% RH), and at least 5 samples of each material were tested, and their values averaged.

#### 2.3.2. Morphology

The morphology was observed using field emission scanning electron microscopy (FESEM) using a ZEISS ULTRA 55 microscope from Oxford Instruments (Abingdon, UK). Before placing the samples in the vacuum chamber, the samples were sputtered with a gold-palladium alloy in an EMITECH sputter coating model SC7620 from Quorum Technologies, Ltd. (East Sussex, UK) applying an acceleration voltage of 2 kV. The fractured surfaces observed corresponded to samples fractured by impact test.

#### 2.3.3. Thermal Analysis

Samples were analyzed by differential scanning calorimetry (DSC) in a Mettler-Toledo 821 calorimeter (Schwerzenbach, Switzerland). Samples with an average weight of approx. 3 mg were subjected to three thermal cycles as follows: (1) heating from 30 °C to 300 °C; (2) cooling from 300 °C to −20 °C; (3) heating from −20 °C up to 330 °C. Heating and cooling rates were set to 10 °C·min^−1^. All tests were run in nitrogen atmosphere with a flow rate of 66 mL·min^−1^ using standard sealed aluminum crucibles (40 µL). The degree of crystallinity ($\chi_{c}$) was determined with the following Equation:(1)$$\chi_{c}(\%)=\left[\frac{{∆H}_{m}}{{∆H}_{100\%}\cdot \omega }\right]\cdot 100$$
where ∆H_m_ (J·g^−1^) stands for the melting enthalpy of the sample, ∆H_100%_ (J·g^−1^) represents the theoretical melting enthalpy of a fully crystalline Polyamide 6 (230 J·g^−1^), and *w* corresponds to the weight fraction of polyamide present in the composite [30].

Thermogravimetric analysis (TGA) was performed in a LINSEIS TGA 1000 (Selb, Germany). Samples with an average weight of approx. 6 mg were placed in standard alumina crucibles of 70 µL and subjected to a heating program from 30 °C to 700 °C at a heating rate of 10 °C·min^−1^ in air atmosphere. The first-derivative thermogravimetry (DTG) curves were also determined, expressing the weight loss rate with the time.

#### 2.3.4. Thermomechanical Characterization

Thermomechanical properties of composites were obtained by dynamical mechanical thermal analyzer DMA1 from Mettler-Toledo (Schwerzenbach, Switzerland), working in single cantilever flexural conditions. Injection-molded samples with dimensions of 20 × 6 × 2.7 mm^3^ were subjected to a dynamic temperature sweep from −100 °C to 150 °C at a constant heating rate of 2 °C·min^−1^. The selected frequency was 1 Hz, and the maximum flexural deformation or deflection was set to 10 µm.

#### 2.3.5. Wetting Characterization

Contact angle measurements were carried out with an EasyDrop Standard goniometer model FM140 (KRÜSS GmbH, Hamburg, Germany), which was equipped with a video capture kit and analysis software (Drop Shape Analysis SW21; DSA1 Version 1.90.0.14). Double distilled water was used as test liquid, with five drops of it put in each sample, and contact angle measurements were taken at 20 s after administration of the water.

#### 2.3.6. Water Uptake Characterization

Injection-molded samples of 4 × 10 × 80 mm^3^ were used. The samples were immersed in distilled water at 23 ± 1 °C. The samples were taken out and, after removing the residual water with a dry cloth, weighed weekly using an analytical balance model AG245 from Mettler Toledo Inc. with a precision of ± 0.1 mg. The evolution of the water absorption was followed for a period of 12 weeks. All measurements were performed in triplicate. The total absorbed water (∆m_t_) during water immersion was calculated following Equation (2):(2)$$∆m_{t}\left(\%\right)=\left(\frac{W-W_{0}}{W_{0}}\right)\times 100$$
where W_0_ is the initial weight and W is the sample weight after an immersion time t of the dry sample before immersion.

## 3. Results

### 3.1. Mechanical Properties of PA-Textile Waste Composites

The mechanical characterization of polyamide composites with different percentages of textile waste provided relevant information on the capabilities and possible applications of the developed composites. Figure 2, Figure 3 and Figure 4 show the main mechanical properties obtained from the different tests performed. With respect to tensile strength (Figure 2), polyamide has a value of 52.6 MPa. The incorporation of 15 wt.% of textile waste causes a slight decrease in the strength of the material, which is less pronounced when silanes are incorporated as coupling agent. In this sense, the effect of silanes is twofold: on the one hand, they mitigate the decrease in the strength of the material, while on the other hand, the silanol groups formed can condense with each other or with hydroxyl groups present on the fiber surface, forming siloxane bonds while improving the dispersion of fibers in the polymer matrix, preventing agglomeration and ensuring a uniform distribution of fibers within the composite [31,32].

For higher amounts of textile waste, the tensile strength increases considerably, reaching 64.5 MPa for 60 wt.% of textile waste. This evidences that the addition of silanes allows the incorporation of higher amounts of textile waste without compromising the strength of the composite material, due to their effect of improving the adhesion between matrix and reinforcement and to the fact that the textile waste is mainly formed by cotton, which, due to its nature, has high tensile strength values [33,34]. Regarding Young’s modulus, as we can see, the polyamide has a value of 582 MPa. It can be observed that the stiffness of the composite material varies as a function of the textile waste content. Thus, by adding 15 wt.% of textile waste, the material becomes stiffer, increasing the elastic modulus. It is also observed that the addition of silanes as a coupling agent further increases this value. This upward trend is also true for higher increments of textile waste percentage in the composite, increasing the Young’s modulus for 30 wt.% of textile waste. From this point, for higher percentages of textile waste, the composite loses stiffness due to the saturation in the concentration of textile waste, which offers greater flexibility, typical of the fibers that make up the composite reinforcement.

With respect to the elongation at break (Figure 3), we can see that the unreinforced polyamide presents values higher than 100%. This incredible ductility is greatly reduced, even with the addition of only 15 wt.% of textile waste, decreasing this value to around 8% elongation. This decrease in the ductility of the polyamide is not solved with the addition of silanes, which do cause a slight improvement in elongation, but in this case, it is not remarkable in comparison with the initial values of the polyamide. The addition of higher amounts of textile waste further decreases the elongation capacity of the composite. Regarding impact strength, polyamide does not show excessively high values. The addition of textile waste increases the brittleness of the material, but the presence of the coupling agent slightly improves the impact resistance, going from 3.78 to 4.40 kJ/m^2^ for 15 wt.% of textile waste with silanes. This behavior shows that the coupling agent used reduces the voids in the matrix–reinforcement interface, which are so sensitive to crack propagation due to the stress concentration effect.

Finally, with respect to the hardness of the different samples analyzed (Figure 4), slight changes in Shore D hardness were observed as the incorporated amount of textile waste increased, but no differences were found due to the inclusion of a compatibilizing agent in the composite.

### 3.2. Morphology of Fractured Surfaces of PA-Textile Waste Composites

The morphology of the components of a composite is directly related to its mechanical performance, since the interface interaction between the polymeric matrix and the reinforcement material determines the main properties of the composite. The analysis of the fractured surfaces obtained after the impact test can provide interesting information to determine the cohesion between the polyamide polymeric matrix and the textile waste filler. Figure 5 shows obtained FESEM images of the impact-fractured surfaces of PA-Textile waste composites.

Figure 5a shows the surface of the polyamide, which has a homogeneous appearance without pronounced roughness and with the formation of crack fronts typical of a polymer with high impact strength. Figure 5b shows the surface of the polyamide sample with 15 wt.% textile waste without the use of silane as coupling agent. The large void (red circle) observed between the textile fiber and the polyamide matrix evidences the lack of cohesion, which translates into poor mechanical properties, especially in terms of impact strength. The incorporation of silanes causes a clear reduction of the separation between the textile resin fiber and the polymeric matrix around it, as shown in Figure 5c. This increase in cohesion is strictly related to an improvement in impact strength. With higher amounts of textile waste incorporated into the composite, a higher amount of textile fiber particles is observed, but in all cases, good adhesion with the polyamide matrix is observed, as can be seen in Figure 5d–f for composites with 30, 45 and 60 wt.% textile waste, respectively. These images validate the adhesion improvement in the interface produced by the silane incorporated in these composites, which improved the mechanical performance of the composites compared to those that did not incorporate it.

### 3.3. Thermal Properties of PA-Textile Waste Composites

Figure 6 shows the DSC thermograms obtained during the second heating cycle of the PA-textile waste composites. The obtained curves for all samples show an endothermic peak indicative of the melting point. Table 3 reports the data related to melting temperature, normalized melting enthalpy and percentage of crystallinity obtained. PA presents a melting temperature of 219.2 °C with a crystallinity value of 28.5%. These values are similar to those obtained by other authors [35,36,37]. Regarding composites with textile waste, no significant differences were observed in the melting temperatures of the resulting composites, with values close to 219 °C. This limited variation is due to the dominant influence of the PA6 matrix. Although the textile waste, primarily composed of cotton and synthetic fibers, has different thermal degradation temperatures, their contribution does not significantly alter the overall thermal stability of the composite.

However, an interesting trend was observed in the crystallinity, which showed a decrease as the textile waste content increased. This trend is similar to that observed in other composites of thermoplastic matrices with lignocellulosic fibers. This is due to the fact that the addition of textile waste reduces the nucleation process and the space available for its growth [38]. It is important to note the significant increase in crystallinity offered by the composite when using silanes as compatibilizer, increasing from 28.5% to 34.2% for 15 wt.% of textile waste. This is due to the effect of the silane acting as a nucleation promoter [39,40]. Finally, the presence of a second, very small melting peak at a temperature close to 260 °C could be detected and was also more noticeable as the textile waste content in the composite increased. This peak was due to the presence of polyester in the textile waste itself.

Regarding the thermal stability of the studied composites, Figure 7 shows the thermogravimetric curves and their derivatives (DTG) for the different compounds. In addition, Table 4 summarizes the main degradation values obtained. As can be seen, the decomposition of polyamide occurs in two steps: the main degradation occurs in the first step between 350 °C and 500 °C, where approximately 85% of mass is lost, and has its maximum degradation temperature around 459 °C. It has been indicated in the literature that the degradation it undergoes in this step corresponds to a β-C-H transfer mechanism, where ketoamides are generated as the main decomposition product [41]. In the second step, which goes from 500 °C to 600 °C, there is a 15% mass loss that can be attributed to the oxidation of the transient residues [42].

Regarding composites with textile waste, it can be observed the incorporation of textile waste produces a decrease in the degradation onset temperature; this is mainly due to the presence of cotton as a major element in the textile waste, whose component cellulose and hemicellulose decomposition occurs at temperatures close to 250 °C and 350 °C, respectively [43]. This fact causes T_5%_ values to decrease as the concentration of textile waste increases, although the incorporation of silanes reduces this effect. T_deg_ values also decrease significantly as textile waste is incorporated into the polyamide, thus affecting the durability of the composite material, and may cause faster degradation of the material over time when exposed to heat. Notable was the presence of two peaks more clearly visible in the DTG curve of the composites with higher textile waste content. These peaks are associated with two degradation processes of the components present in the textile waste. The first one, called the second pyrolysis of cotton, was due to the decomposition of lignin at temperatures above 400 °C [35]. The second one was due to the degradation of the polyester present in the textile waste, which also occurred in the temperature range close to 400 °C.

With respect to residual mass results, the introduction of textile waste particles hardly changed the residual mass values of the composites, because the main component of the textile waste was completely decomposed before 700 °C.

### 3.4. Thermomechanical Properties of PA-Textile Waste Composites

Figure 8 and Table 5 show the thermodynamic curves and values respectively obtained with DMTA of the PA composites with textile waste. Figure 8a shows the evolution of the storage modulus (G′) between the temperatures of −100 °C and 150 °C. As expected, the general trend in all the samples analyzed is a decrease in G′ as temperature increases. Polyamide presents a G′ at −100 °C of 1612 MPa, and as the textile waste is incorporated, the G′ increases until reaching a maximum value of 2437 MPa for a content of 60 wt.%. The increase in stiffness of the composites as the textile residue content increases is evident as the textile residue content increases. In addition, the results corroborate the compatibilizing character in the composite of 15 wt.% of textile waste with silanes with respect to the one that does not incorporate it. Figure 8b shows the evolution of the dynamic damping factor (tan δ) with temperature. Polyamide presents a peak close to −67 °C, corresponding to the γ relaxation and related to local movements of the side chains of the polyamide main chain; as can be observed, the peak related to γ relaxation is reduced as textile waste is incorporated [44,45]. On the other hand, the glass transition T_g_ (visualized as the α-relaxation) has a maximum of the tan δ at around 40 °C and involves the cooperative movement of long sections of the polymer chain. The intensity and position of α-relaxation can change due to the filler–matrix molecular interactions. Regarding the composites, no significant changes were observed in the glass transition temperature for composites at low filler content. However, the peak related to α-relaxation partially disappears by incorporating textile waste. This can indicate a better interaction between the fiber and the matrix promoted by the addition of silanes [46]. In addition, the slight increase in T_g_ indicates an increase in stiffness as the amount of incorporated textile waste increases.

### 3.5. Wetting Properties of PA-Textile Waste Composites

In order to evaluate the behavior of the composites toward water, the contact angle was measured at different times after applying a drop of water on the surface of each sample. A high contact angle value is indicative of low affinity to water (hydrophobicity), while a high contact angle value is indicative of high hydrophilicity and consequently better adhesion ability. Table 6 shows the water contact angles on surface composites analyzed. It can be seen that polyamide reaches a contact angle value of 66.8, which means that we are dealing with a hydrophilic material. This confirms the wetting character of polyamide due to the presence of polar amide groups, which tend to react with water. With regard to the composites, a decreasing trend can be clearly observed in the contact angle as the textile waste content increases. The minimum value of 56.1 for the contact angle is obtained for a textile waste content of 60 wt.%, which represents a decrease of 16.1% of contact angle. This means that the hydrophilic character of the textile is still superior to that of the polyamide due its composition described in Table 1, with the presence of strongly hydrophilic fibers such as cotton with high wettability [47]. Such behavior can be attributed to the action of the hydroxyl and carbonyl groups present in the cellulose, hemicellulose and lignin that make up the cotton fibers. These groups give polarity to the composite and can form hydrogen bonds with water molecules, which is a polar solvent [25]. With regard to the addition of silanes, a slight decrease in the hydrophilic character of the mixture can be observed, which may be due to a better interaction between the polyamide and textile waste bonds, slightly overlapping the hydrophilic character of cotton, which is still superior to that of polyamide.

### 3.6. Water Uptake Characterization of PA-Textile Waste Composites

The water absorption test is closely related to the wetting properties as described above, but in this case, we are studying the capacity to absorb water over a prolonged period of time. For this purpose, water immersion of each of the samples was studied for 11 weeks. The results can be observed in Figure 9. As can been observed, the results again show a completely hydrophilic character of the composites, which corroborates the results obtained in the goniometry test. In this case, a concave down increasing curve of water absorption can be observed for all the samples, since during the first weeks, the absorption they present is very large and decreases as time goes by, stabilizing around week 11 for most of the samples. In addition, it is again observed that the hydrophilic character of the textile waste is superior to that of the polyamide, confirming that water absorption is higher as the composite has more textile waste content. However, after 8 weeks, an interesting phenomenon occurs, since it is observed how the samples with higher percentages of textile waste begin to stabilize their water absorption, reaching saturation between 10 and 11 weeks, with an absorbed water value of around 9.5%, highlighting the potential long-term consequences for the structural integrity and performance of the composites. These results are very similar to those for composites with low textile waste content and even similar to polyamide itself without textile waste. This behavior is due to what was mentioned in the previous section, since water produces polar bonds with the amide group because the nitrogen and oxygen atoms in the amide groups have partial negative charges, which attract the partial positive charge of the hydrogen atoms in water molecules. In addition the small water molecules occupy spaces between the polyamide molecules.

### 3.7. Visual Aspect of PA-Textile Waste Composites

Figure 10 shows a photograph of the injection-molded pieces of PA-textile waste composites. As can be seen, the polyamide without textile waste content shows a white color typical of a virgin plastic without dyes. The introduction of textile waste even in low quantities produces a very radical change in the color of the composite, offering a shade close to dark brown in all its compositions. The dark brown appearance of the composites is primarily due to the inherent color of the textile waste materials used, which can include natural and synthetic fibers with various dyes and impurities that darken during processing. The composite samples with higher textile waste content display some brown splay marks caused by tiny gas bubbles that are dragged across the surface of the sample when the mold cavity is filled [48], being in this case more difficult to obtain a homogeneous color. This fact can detract from the visual appeal and may limit the use of these composites in high-end or visually sensitive applications.

## 4. Conclusions

The results obtained in this work allow us to validate the incorporation of textile waste into polyamide to obtain composites at a relatively low cost. The main achievements of the work are the following:The use of very abundant and low-cost reinforcing fillers such as textile waste can increase the performance of the material obtained.The introduction of silane coupling agents significantly improved the adhesion between the polyamide matrix and textile waste fibers. This enhanced compatibility resulted in better mechanical properties of the composites, making the material suitable for various applications.The mechanical characterization revealed that incorporating up to 60 wt.% of textile waste into the polyamide matrix did not compromise the material’s strength. Instead, it enhanced tensile strength, particularly when silanes were used as coupling agents. This indicates that high levels of textile waste can be successfully utilized without degrading the composite’s performance.The incorporation of textile waste into the PA matrix decreases the degradation onset temperature due to the presence of cotton; in addition, silane coupling agents help mitigate the reduction in degradation onset temperature, improving thermal stability.The incorporation of textile waste into the PA matrix affects the surface properties and thus the wettability of the composites.The hydrophilic character of the textile waste, especially cotton, was confirmed as superior to that of polyamide. This was evidenced by the higher water absorption in composites with greater textile waste content.Regarding the visual aspect of the composites, those with higher textile waste content exhibited some brown splay marks on their surfaces.

In conclusion, the utilization of textile waste in polymer composites presents an environmentally friendly solution to textile waste management, contributing to sustainability in the textile and polymer industries. This approach not only reduces waste but also provides a cost-effective method for producing high-performance composite materials.

## Figures and Tables

**Figure 1 polymers-16-02061-f001:**
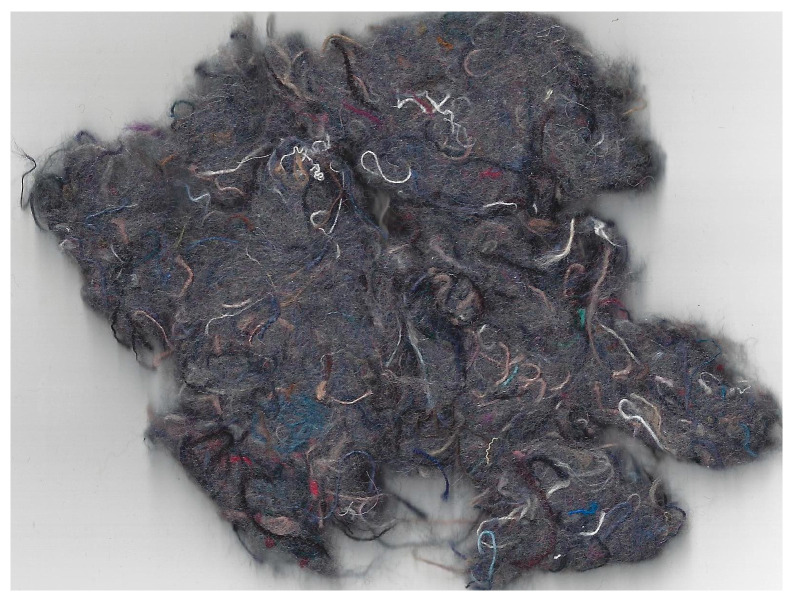
Visual aspect of the used textile waste from post-consumption of standard garments made with woven fabric.

**Figure 2 polymers-16-02061-f002:**
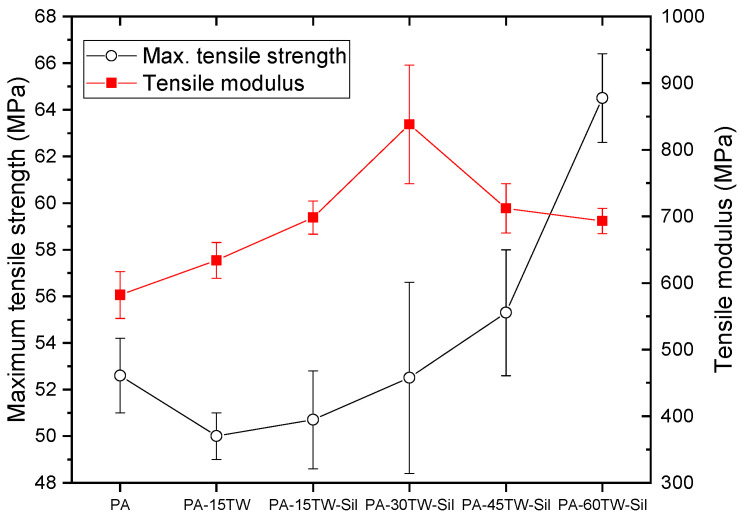
Results obtained from mechanical characterization of the injection-molded samples of PA-Textile waste composites in terms of maximum tensile strength and tensile modulus.

**Figure 3 polymers-16-02061-f003:**
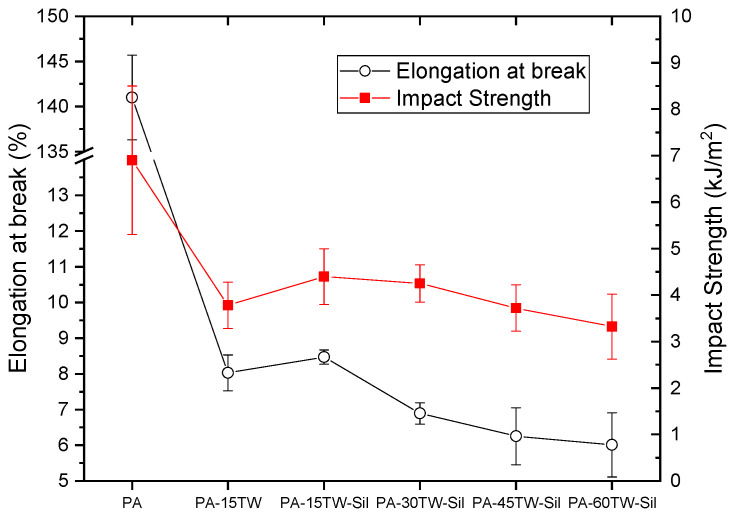
Results obtained from mechanical characterization of the injection-molded samples of PA-Textile waste composites in terms of elongation at break and impact strength.

**Figure 4 polymers-16-02061-f004:**
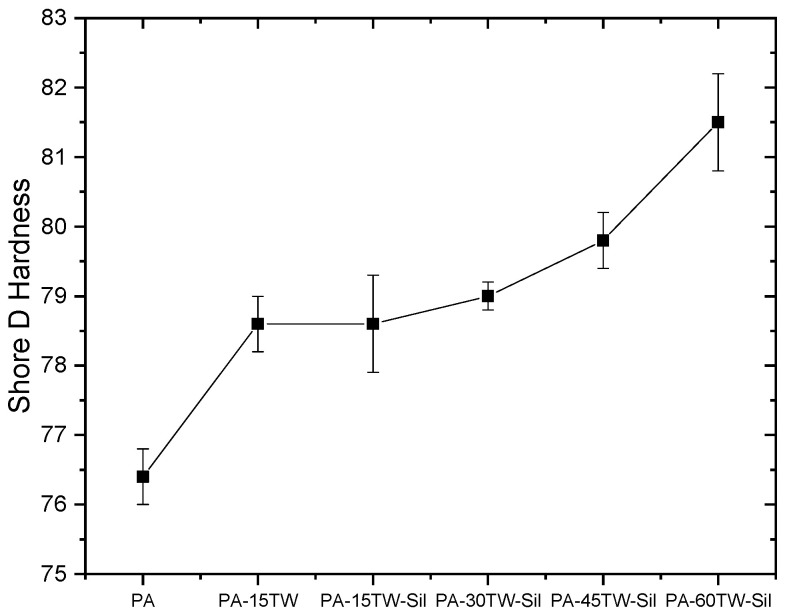
Results obtained from mechanical characterization of the injection-molded samples of PA-Textile waste composites in terms of Shore D hardness.

**Figure 5 polymers-16-02061-f005:**
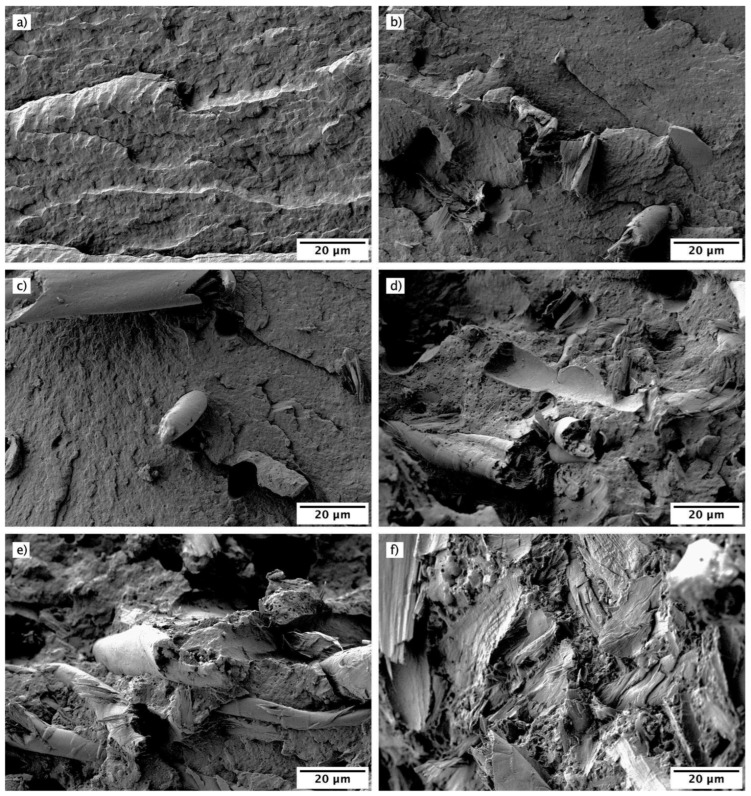
Field emission scanning electron microscopy (FESEM) images at 1000× of the impact-fractured surfaces of PA-textile waste composites: (**a**) PA; (**b**) PA-15TW; (**c**) PA-15TW-Sil; (**d**) PA-30TW-Sil; (**e**) PA-45TW-Sil; (**f**) PA-60TW-Sil.

**Figure 6 polymers-16-02061-f006:**
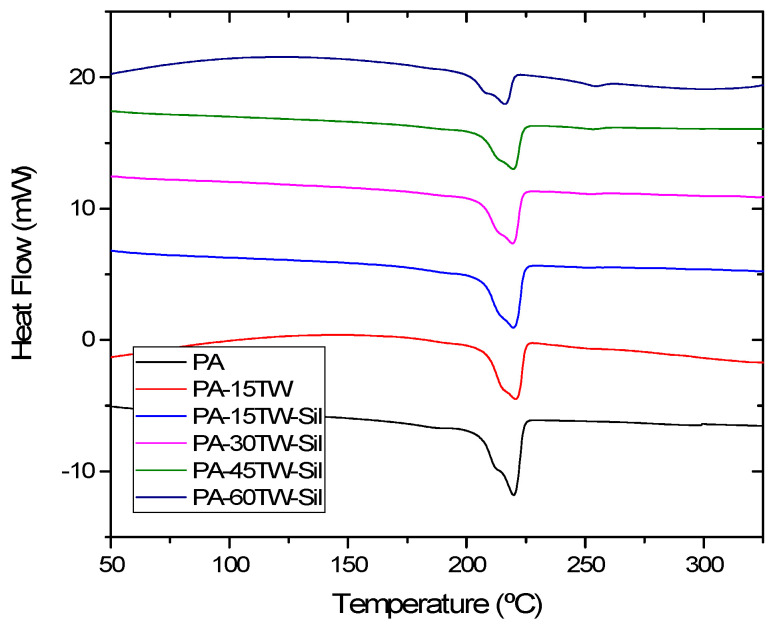
Differential scanning calorimetry (DSC) thermograms of PA-textile waste composites.

**Figure 7 polymers-16-02061-f007:**
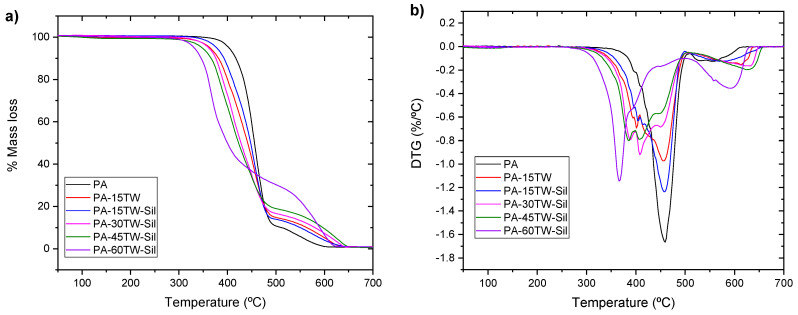
TGA curves of PA-textile waste composites: (**a**) thermogravimetric analysis (TGA) curves; (**b**) first derivative (DTG).

**Figure 8 polymers-16-02061-f008:**
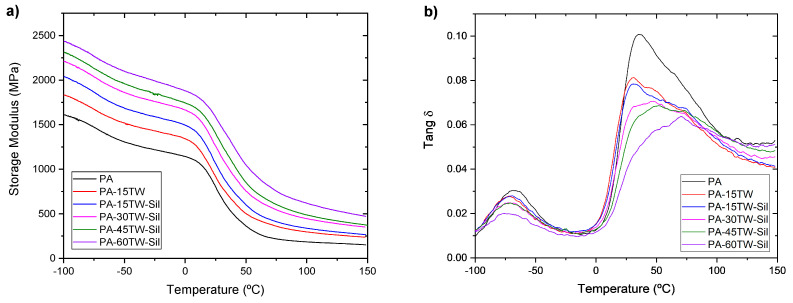
DMTA curves of PA-textile waste composites: (**a**) storage modulus (G′) and (**b**) dynamic damping factor (tan δ).

**Figure 9 polymers-16-02061-f009:**
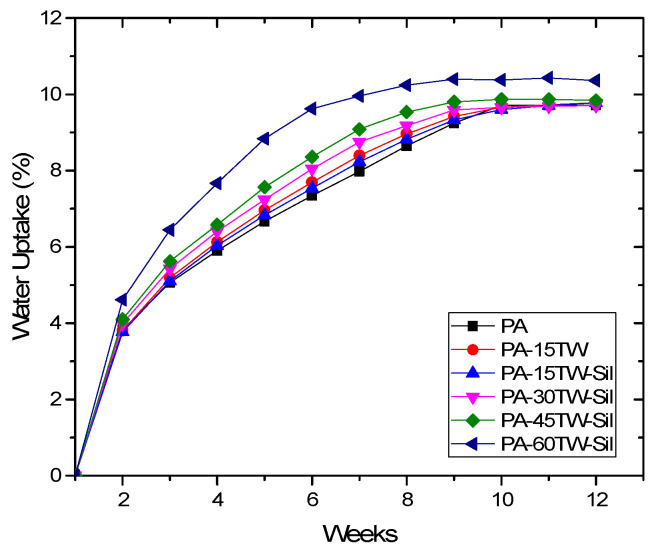
Water uptake of the injection-molded pieces made of PA-textile waste composites.

**Figure 10 polymers-16-02061-f010:**
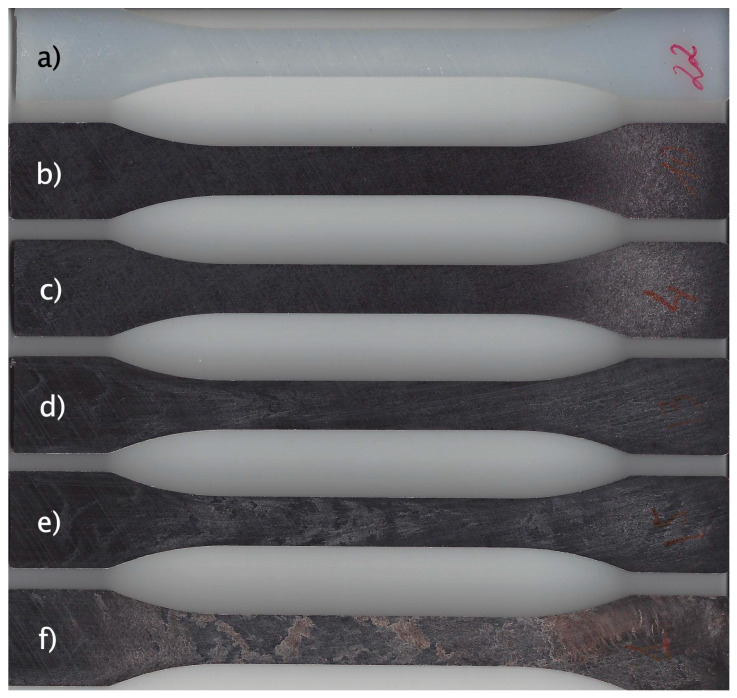
Visual appearance of injection-molded pieces of PA-textile waste composites: (**a**) PA; (**b**) PA-15TW; (**c**) PA-15TW-Sil; (**d**) PA-30TW-Sil; (**e**) PA-45TW-Sil; (**f**) PA-60TW-Sil.

**Table 1 polymers-16-02061-t001:** Composition of textile waste used from post-consumption of standard garments made with woven fabric.

Material	Composition (wt.%)
Cotton	71.5
Polyester	16.2
Viscose	4.9
Polyamide	3.3
Wool	2.8
Acrylic	1.3

**Table 2 polymers-16-02061-t002:** Summary of compositions according to the weight content (wt.%) of PA-textile waste composites.

Code	PA (wt.%)	Textile Waste (wt.%)	Silane (wt.%)
PA	100	0	0
PA-15TW	85	15	0
PA-15TW-Sil	84	15	1
PA-30TW-Sil	69	30	1
PA-45TW-Sil	54	45	1
PA-60TW-Sil	39	60	1

**Table 3 polymers-16-02061-t003:** Main thermal parameters of the of PA-textile waste composites in terms of melting temperature (T_m_), normalized melting enthalpy (∆H_m_), and percentage of crystallinity (χ_c_).

Code	T_m_ (°C)	∆H_m_ (J/g)	χ_c_ (%)
PA	219.2 ± 1.1	65.6 ± 1.1	28.5 ± 1.1
PA-15TW	220.4 ± 1.2	52.5 ± 1.0	26.8 ± 1.1
PA-15TW-Sil	219.1 ± 0.9	66.1 ± 0.9	34.2 ± 0.8
PA-30TW-Sil	218.9 ± 0.8	47.8 ± 0.8	30.1 ± 0.8
PA-45TW-Sil	219.3 ± 1.2	34.6 ± 1.1	27.9 ± 1.0
PA-60TW-Sil	218.8 ± 1.0	22.6 ± 1.2	25.2 ± 1.2

**Table 4 polymers-16-02061-t004:** Main thermal degradation parameters of the samples of PA-textile waste composites in terms of temperature at mass loss of 5% (T_5%_), maximum degradation rate (peak) temperature (T_deg_), and residual weight at 700 °C.

Code	T_5%_ (°C)	T_deg_ (°C)	Residual Mass (%)
PA	403 ± 1.1	459 ± 1.1	0.62 ± 0.1
PA-15TW	364 ± 1.2	457 ± 1.0	0.72 ± 0.2
PA-15TW-Sil	378 ± 0.9	459 ± 0.7	0.64 ± 0.3
PA-30TW-Sil	361 ± 0.8	408 ± 0.8	0.66 ± 0.5
PA-45TW-Sil	347 ± 1.2	385 ± 1.2	0.78 ± 0.5
PA-60TW-Sil	333 ± 1.0	366 ± 1.0	0.96 ± 0.3

**Table 5 polymers-16-02061-t005:** Main thermomechanical parameters of PA-textile waste composites.

Code	G′ (MPa) at −150 °C	G′ (MPa) at 0 °C	G′ (MPa) at 60 °C	T_g_ (°C)
PA	1612 ± 11	1143 ± 12	277 ± 8	40.0 ± 0.7
PA-15TW	1832 ± 13	1342 ± 18	430 ± 5	38.2 ± 0.9
PA-15TW-Sil	2036 ± 17	1495 ± 17	496 ± 5	38.9 ± 1.3
PA-30TW-Sil	2211 ± 22	1664 ± 21	649 ± 9	39.2 ± 1.1
PA-45TW-Sil	2311 ± 23	1743 ± 15	723 ± 7	40.5 ± 1.3
PA-60TW-Sil	2437 ± 18	1880 ± 18	906 ± 8	41.8 ± 1.1

**Table 6 polymers-16-02061-t006:** Water contact angle values for PA-textile waste composites.

Code	Contact Angle (°)
PA	66.8 ± 1.3
PA-15TW	65.4 ± 1.1
PA-15TW-Sil	66.2 ± 0.6
PA-30TW-Sil	64.9 ± 1.6
PA-45TW-Sil	60.5 ± 1.9
PA-60TW-Sil	56.1 ± 1.3

## Data Availability

The original contributions presented in the study are included in the article, further inquiries can be directed to the corresponding author/s.

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
