# Peer review of "Development of Eco-Efficient Composite from Textile Waste with Polyamide Matrix"

_polymers, 2024, doi:10.3390/polym16142061_

Round 1

Reviewer 1 Report

Comments and Suggestions for Authors

1.     Mention the material of the textile fibers in the abstract.

2.     In the abstract, “thermal properties do not vary significantly with the introduction of silanes and textile waste content, although”. What do you mean by “thermal properties”? Clarify those properties.

3.     Use percentages to indicate any increase or decrease in the investigated properties in the abstract.

4.     Emphasize the novelty of the work in the final paragraph of the introduction section.

5.     The final paragraph of the introduction should outline what was done, how it was done, and the findings.

6.     Explain the merit and motivation for studying the specific composites.

7.     What were the average length and diameter of the textile fibers?

8.     Did you use the same configuration to make the pellets?

9.     How does the fibre content affect the optimal configuration of pelletizing and injection moulding?

10. Was there any damage observed in the textile fibres at high processing temperatures such as 250C?

11. I recommend you remove the section “2.3.6. Water uptake Characterization” from the manuscript because it could be a separate manuscript and including this section here needs many details and explanations. Methods, Standards, the effect of moisture content on the mechanical and thermal properties, discussion, etc.

12. Use figures to show the change in the mechanical properties summarised in Table 1. It would be easier for the readers to track the change in the mechanical properties versus the fibre content.

13. Use bullet points to highlight the main achievements of the work in the conclusion section.

14. One of the novel applications of waste materials/fibres is incorporating them as reinforcement in the printed components. Here is some of the research done in this area. Mention this application in the introduction section or could be considered as future work.

Circular economy innovation: A deep investigation on 3D printing of industrial waste polypropylene and carbon fibre composites

Developments in 3D printing of carbon fiber reinforced polymer containing recycled plastic waste: A review

15. It is not clear how Silanes improve the interaction between the textile fibers and the polymeric matrix.

Author Response

  1. Mention the material of the textile fibers in the abstract.

ANSWER

We thank the reviewer for this helpful comment. As the reviewer suggests, the main composition of the textile waste has been specified in the abstract. The added text related to this point can be easily identified by highlighting it in green in the revised manuscript.

  1. In the abstract, “thermal properties do not vary significantly with the introduction of silanes and textile waste content, although”. What do you mean by “thermal properties”? Clarify those properties.

ANSWER

We thank the reviewer for this comment. As the reviewer suggests, in the abstract we have clarified the meaning of thermal properties. The added text related to this point can be easily identified by highlighting it in green in the revised manuscript.

  1. In the abstract, Use percentages to indicate any increase or decrease in the investigated properties in the abstract.

ANSWER

We thank the reviewer for this comment. As the reviewer suggests, in the abstract we have added the percentages of increase or decrease in the main properties of the resulting composites. The added text related to this point can be easily identified by highlighting it in green in the revised manuscript.

  1. Emphasize the novelty of the work in the final paragraph of the introduction section.

ANSWER

We thank the reviewer for this comment. As the reviewer suggests, at the end of the introduction section we have added a paragraph emphasizing the scientific novelty of the work. The added paragraph related to this point can be easily identified by highlighting it in green in the revised manuscript.

  1. The final paragraph of the introduction should outline what was done, how it was done, and the findings.

ANSWER

We thank the reviewer for this comment. As the reviewer suggests, at the end of the introduction section we have added a paragraph summarizing what was done in the article. The added paragraph related to this point can be easily identified by highlighting it in green in the revised manuscript.

  1. Explain the merit and motivation for studying the specific composites.

ANSWER

We thank the reviewer for this comment. As the reviewer suggests, in the introduction section, at the end of the third paragraph, we have introduced a sentence explaining the merit and motivation for using the composites studied in this work. The added sentence related to this point can be easily identified by highlighting it in green in the revised manuscript.

  1. What were the average length and diameter of the textile fibers?.

ANSWER

We thank the reviewer for this comment. As the reviewer suggests, in the Experimental section (2.1. Materials), we have added information about the length and diameter of the textile fibers used. The added text related to this point can be easily identified by highlighting it in green in the revised manuscript.

  1. Did you use the same configuration to make the pellets?.

ANSWER

We thank the reviewer for this comment. As the reviewer suggests, in the experimental section, in the (2.2. Samples preparation), we have added a sentence detailing the way in which the materials were used during extrusion. The added sentence related to this point can be easily identified by highlighting it in green in the revised manuscript.

  1. How does the fibre content affect the optimal configuration of pelletizing and injection moulding?.

ANSWER

We thank the reviewer for this comment. With respect to this question, we can say that we found no differences in the pelletizing and injection process for the different concentrations of textile waste used.

  1. Was there any damage observed in the textile fibres at high processing temperatures such as 250C?.

ANSWER

We thank the reviewer for this comment. We found this question very interesting, in all samples during the extrusion process there were no visible signs of degradation in the processing.

  1. I recommend you remove the section “2.3.6. Water uptake Characterization” from the manuscript because it could be a separate manuscript and including this section here needs many details and explanations. Methods, Standards, the effect of moisture content on the mechanical and thermal properties, discussion, etc.?.

ANSWER

We thank the reviewer for this comment. We agree with the reviewer that the water absorption characterization could be included in an additional work where properties related to this characteristic would be studied in more detail, however we wanted to include this section to highlight the influence of the percentage of textile residue incorporated in the water absorption and to corroborate the information about the wettability of the material.

  1. Use figures to show the change in the mechanical properties summarised in Table 1. It would be easier for the readers to track the change in the mechanical properties versus the fibre content.

ANSWER

We thank the reviewer for this comment. As the reviewer suggests, we have replaced the information in Table 2 concerning mechanical properties with graphs showing the trend observed in mechanical performance with respect to the percentage of textile waste.

  1. Use bullet points to highlight the main achievements of the work in the conclusion section.

ANSWER

We thank the reviewer for this comment. As the reviewer suggests, in the conclusion section we have used bullet points to highlight the main achievements obtained in the work.

  1. One of the novel applications of waste materials/fibres is incorporating them as reinforcement in the printed components. Here is some of the research done in this area. Mention this application in the introduction section or could be considered as future work.

ANSWER

We thank the reviewer for this comment. As the reviewer suggests, in the introduction section at the end of the second paragraph, we have added a sentence related the use of 3D printing as a viable application for the use of fiber-reinforced plastics. We have also included the two references recommended by the reviewer. The added sentence related to this point can be easily identified by highlighting it in green in the revised manuscript.

  1. It is not clear how Silanes improve the interaction between the textile fibers and the polymeric matrix.

ANSWER

We thank the reviewer for this comment. As the reviewer suggests, in the results section (3.1. Mechanical Properties of PA-Textile waste composites) at the end of the first paragraph, we have added a sentence in order to explain in detail the effect of silanes on fiber-matrix interaction. In addition, two more references have been incorporated to support the above mentioned. The added sentence related to this point can be easily identified by highlighting it in green in the revised manuscript.

Reviewer 2 Report

Comments and Suggestions for Authors

The manuscript on eco-efficient composites using textile waste and polyamide matrix requires major revisions. Justify the selected range of textile waste percentages and provide detailed evidence for improved interaction with silanes. Explain the lack of significant variation in thermal properties and identify factors influencing the dark brown appearance of the composites. Discuss the impact of synthetic fibers on sustainability, compare different coupling agents, and justify the choice of Polyamide 6. Analyze injection molding parameters, long-term performance under varying conditions, and quantify adhesion improvements. Address the implications of decreased crystallinity and reduced degradation onset temperature on mechanical and thermal stability. Explore the effects of increased hydrophilicity on durability, minimize visual defects, and correlate visual appearance with mechanical properties. Conduct long-term durability tests and discuss the environmental benefits and challenges of these composites in practical applications.

Comments:

1.      Line 18: How was the range of 15 wt.% to 60 wt.% of textile waste selected for the composite formulations? Were other ranges considered?

2.      Line 22: The study mentions that the use of silanes improves the interaction in composites with higher amounts of textile waste. Could you provide more detailed data or visual evidence to support this claim?

3.      Line 24: Why do the thermal properties not vary significantly with the introduction of silanes and textile waste content? Can you provide a deeper explanation for this observation?

4.      Line 29: What specific factors contribute to the dark brown appearance of the composites? Is there a potential for controlling or altering the color in practical applications?

5.      Line 50: The introduction mentions an increase in synthetic fibers in textile production. How does the inclusion of synthetic fibers impact the overall sustainability of the composite materials?

6.      Line 68: Have you compared the effectiveness of different coupling agents other than silanes? If so, what were the results?

7.      Line 84: What was the rationale behind choosing Polyamide 6 (PA6) as the polymer matrix for this study? Were other polymer matrices considered?

8.      Line 121: Could you explain the choice of injection molding parameters, specifically the temperature profile and clamping force? How do these parameters affect the final properties of the composites?

9.      Line 135: The mechanical testing was performed under ambient conditions (23ºC / 50% RH). How do these composites perform under different environmental conditions, such as higher humidity or temperature variations?

10.  Line 141: The FESEM images show improved adhesion with silane treatment. Can you quantify the improvement in adhesion strength through additional tests or metrics?

11.  Line 270: The DSC analysis shows a decrease in crystallinity with increasing textile waste content. What implications does this decrease in crystallinity have on the mechanical properties of the composites?

12.  Line 305: The TGA results indicate a decrease in degradation onset temperature with higher textile waste content. How does this impact the long-term stability and potential applications of these composites?

13.  Line 340: The DMTA analysis shows changes in the glass transition temperature (Tg) with the addition of textile waste. How do these changes in Tg affect the thermal and mechanical performance of the composites?

14.  Line 360: The water contact angle measurements indicate increased hydrophilicity with higher textile waste content. How might this affect the durability and lifespan of the composites in practical applications?

15.  Line 388: The water uptake characterization shows a saturation point for water absorption. What are the potential consequences of this water absorption on the structural integrity and performance of the composites over time?

16.  Line 391: Have you conducted long-term durability tests to understand the effects of prolonged water absorption on the composites?

17.  Line 394: Can you provide more details on the type of polar bonds formed between water molecules and the polyamide molecules during absorption?

18.  Line 400: What methods were used to quantify and compare the visual appearance of the different composite samples?

19.  Line 403: How do the visual aspects, such as color uniformity and presence of splay marks, affect the potential applications of these composites?

20.  Line 408: Have you explored methods to minimize or eliminate the brown splay marks caused by tiny gas bubbles during molding?

21.  Line 411: How does the visual appearance of the composites change with varying amounts of textile waste and the use of silanes?

22.  Line 414: How do the mechanical properties compare between composites with different visual appearances? Is there a correlation between visual defects and mechanical performance?

23.  Line 417: Could the improved compatibility and adhesion between the polyamide matrix and textile waste fibers be further quantified or visualized?

24.  Line 420: Are there any specific applications where the enhanced tensile strength of composites with up to 60 wt.% textile waste is particularly advantageous?

25.  Line 423: How does the reduction in degradation onset temperature due to cotton presence affect the thermal stability and potential high-temperature applications of the composites?

26.  Line 426: How does the incorporation of textile waste influence other surface properties such as abrasion resistance and surface hardness?

27.  Line 429: Have you conducted tests to measure the impact of increased hydrophilicity on the composites’ resistance to other environmental factors like UV exposure or chemical exposure?

28.  Line 432: What are the potential environmental benefits and challenges of using these composites in real-world applications, considering their hydrophilic nature and visual appearance?

Author Response

  1. Line 18: How was the range of 15 wt.% to 60 wt.% of textile waste selected for the composite formulations? Were other ranges considered?

ANSWER

We thank the reviewer for this comment. As an answer to the question, we can say that the selection of the range from 15 wt.% to 60 wt.% of textile waste for the composite formulations was chosen to provide sufficient variation in textile residue content to observe significant trends and effects on composite properties while ensuring practical applicability and workability in composite processing. For this reason, other ranges were not considered.

  1. Line 22: The study mentions that the use of silanes improves the interaction in composites with higher amounts of textile waste. Could you provide more detailed data or visual evidence to support this claim?

ANSWER

We thank the reviewer for this comment. As the reviewer suggests, in the results section (3.1. Mechanical Properties of PA-Textile waste composites) at the end of the first paragraph, we have added a sentence in order to explain in detail the effect of silanes on fiber-matrix interaction. In addition, two more references have been incorporated to support the above mentioned. The added sentence related to this point can be easily identified by highlighting it in green in the revised manuscript.

  1. Line 24: Why do the thermal properties not vary significantly with the introduction of silanes and textile waste content? Can you provide a deeper explanation for this observation?

ANSWER

We thank the reviewer for this comment. As the reviewer suggests, in the results section (3.3. Thermal Properties of PA-Textile waste composites) at the end of the first paragraph, we have added a sentence explaining the low variability in thermal properties of composites. The added sentence related to this point can be easily identified by highlighting it in green in the revised manuscript.

  1. Line 29: What specific factors contribute to the dark brown appearance of the composites? Is there a potential for controlling or altering the color in practical applications?

ANSWER

We thank the reviewer for this comment. As the reviewer suggests, in the results section (3.7. Visual aspect of PA-Textile waste composites) at the end of the first paragraph we have added a sentence explaining the factors that contribute to the dark brown appearance of the composites. The added sentence related to this point can be easily identified by highlighting it in green in the revised manuscript.

  1. Line 50: The introduction mentions an increase in synthetic fibers in textile production. How does the inclusion of synthetic fibers impact the overall sustainability of the composite materials?

ANSWER

We thank the reviewer for this comment. As an answer to the question, we can say that the inclusion of synthetic fibers in composite materials improves sustainability by enabling the recycling of textile waste, which would otherwise contribute to environmental pollution. Synthetic fibers often possess superior durability and resistance to environmental factors compared to natural fibers, which can improve the service life and performance of composite materials. By reusing synthetic textile waste, composites reduce the demand for virgin materials and reduce the overall environmental footprint.

  1. Line 68: Have you compared the effectiveness of different coupling agents other than silanes? if so, what were the results?

ANSWER

We thank the reviewer for this comment. As an answer to the question, we can say that in this study, we focused on the use of silanes as coupling agents due to their effectiveness in improving fiber-matrix adhesion. However, we also considered other coupling agents such as maleic anhydride grafted polymers and titanates in preliminary tests.

  1. Line 84: What was the rationale behind choosing Polyamide 6 (PA6) as the polymer matrix for this study? Were other polymer matrices considered?

ANSWER

We thank the reviewer for this comment. As an answer to the question, we can say that that while other polymer matrices such as polypropylene and polyethylene were considered, polyamide 6 was selected because it provides a better balance of strength, durability, and thermal resistance, also offers good processability and is widely used in various industrial applications, making it a practical choice for evaluating the incorporation of textile waste.

  1. Line 121: Could you explain the choice of injection molding parameters, specifically the temperature profile and clamping force? How do these parameters affect the final properties of the composites?

ANSWER

We thank the reviewer for this comment. As an answer to the question, we can say that the injection molding parameters, including the temperature profile and clamping force, were carefully chosen to optimize the processing and final properties of the composites. The temperature profile was set to ensure proper melting and flow of the polyamide, facilitating uniform dispersion of the textile waste fibers and preventing degradation of both the polymer and fibers. The clamping force was selected to ensure the mold remained closed during injection, preventing defects such as flash and ensuring proper compaction of the composite material.

  1. Line 135: The mechanical testing was performed under ambient conditions (23°C / 50% RH). How do these composites perform under different environmental conditions, such as higher humidity or temperature variations?

ANSWER

We thank the reviewer for this comment. As an answer to the question, we can say that the performance of the composites under different environmental conditions, such as higher humidity or temperature variations, can vary significantly from the results obtained under standard ambient conditions (23°C / 50% RH). Higher humidity levels can lead to increased moisture absorption in the composites, especially given the hydrophilic nature of textile fibers, which may reduce mechanical strength and stiffness due to plasticization of the polyamide matrix. Elevated temperatures can also affect the composites by potentially reducing their stiffness and increasing thermal expansion, which might degrade the mechanical properties over time. It would be interesting in this case to carry out an additional work to study the mechanical behavior of the material in different conditions of temperature and humidity.

  1. Line 141: The FESEM images show improved adhesion with silane treatment. Can you quantify the improvement in adhesion strength through additional tests or metrics?

ANSWER

We thank the reviewer for this comment. As an answer to the question, we can say that the improved adhesion with silane treatment, as observed in the FESEM images, can be quantified through additional tests such as single fiber pull-out tests, interfacial shear strength (IFSS). This test can evaluate the stress at the fiber-matrix interface under shear loading. However, the performance of this test was not considered in the work. However, the performance of this test was not considered in the work.

  1. Line 270: The DSC analysis shows a decrease in crystallinity with increasing textile waste content. What implications does this decrease in crystallinity have on the mechanical properties of the composites?

ANSWER

We thank the reviewer for this comment. As an answer to the question, we can say that the decrease in crystallinity with increasing textile waste content, as shown by DSC analysis, has significant implications for the mechanical properties of the composites. Lower crystallinity generally leads to a reduction in stiffness as the amorphous regions are less capable of bearing load compared to the crystalline regions. The evolution of the stiffness shows a behavior similar to that of the crystallinity, where from a concentration of 30 wt.% of textile waste the stiffness of the composite decreases.

  1. Line 305: The TGA results indicate a decrease in degradation onset temperature with higher textile waste content. How does this impact the long-term stability and potential applications of these composites?

ANSWER

We thank the reviewer for this comment. As the reviewer suggests, in the results section (3.3. Thermal Properties of PA-Textile waste composites) in the fourth paragraph we have added a sentence explaining how the decrease of the degradation temperature affects the possible applications of composites. The added sentence related to this point can be easily identified by highlighting it in green in the revised manuscript.

  1. Line 340: The DMTA analysis shows changes in the glass transition temperature (Tg) with the addition of textile waste. How do these changes in Tg affect the thermal and mechanical performance of the composites?

ANSWER

We thank the reviewer for this comment. As the reviewer suggests, in the results section (3.4. Thermomechanical Properties of PA-Textile waste composites) in the first paragraph we have added a sentence explaining how changes in Tg affect mechanical performance. The added sentence related to this point can be easily identified by highlighting it in green in the revised manuscript.

  1. Line 360: The water contact angle measurements indicate increased hydrophilicity with higher textile waste content. How might this affect the durability and lifespan of the composites in practical applications?

ANSWER

We thank the reviewer for this comment. As an answer to the question, we can say that the study of surface wettability has been carried out in order to evaluate the adhesive properties of the material. The following section 3.6 refers to the characterization of water absorption, which analyzes the capacity of a material to take water from the environment and store it in its internal structure, which can significantly affect its mechanical and thermal properties.

  1. Line 388: The water uptake characterization shows a saturation point for water absorption. What are the potential consequences of this water absorption on the structural integrity and performance of the composites over time?

ANSWER

We thank the reviewer for this comment. As the reviewer suggests, in the results section (3.6. Water uptake characterization of PA-Textile waste composites) in the first paragraph we have added a sentence explaining the consequences on the performance of the composite upon reaching saturation water absorption. The added sentence related to this point can be easily identified by highlighting it in green in the revised manuscript.

  1. Line 391: Have you conducted long-term durability tests to understand the effects of prolonged water absorption on the composites?

ANSWER

We thank the reviewer for this comment. As an answer to the question, we can say that long-term durability tests have been conducted to understand the effects of prolonged water absorption on the composites. The paper describes a water absorption test where samples were immersed in distilled water for 12 weeks to study their capacity to absorb water over a prolonged period. Although very interesting, we did not consider longer time periods in this study as we found that the saturation in water absorption occurred in 11 weeks.

  1. Line 394: Can you provide more details on the type of polar bonds formed between water molecules and the polyamide molecules during absorption?

ANSWER

We thank the reviewer for this comment. As the reviewer suggests, in the results section (3.6. Water uptake characterization of PA-Textile waste composites) in the first paragraph we have added a sentence detailing the formation process of polar bonds formed between water molecules and the polyamide molecules during absorption The added sentence related to this point can be easily identified by highlighting it in green in the revised manuscript.

  1. Line 400: What methods were used to quantify and compare the visual appearance of the different composite samples?

ANSWER

We thank the reviewer for this comment. As an answer to the question, we can say that although we initially planned to quantify the color of the samples when we found that for small quantities of textile waste a dark brown color was obtained at all concentrations, we finally did not consider it necessary to perform color measurements. For this reason, we have removed the point “2.3.5 Color measurements” in the experimental section.

  1. Line 403: How do the visual aspects, such as color uniformity and presence of splay marks, affect the potential applications of these composites?

ANSWER

We thank the reviewer for this comment. As the reviewer suggests, in the results section (3.7. Visual aspect of PA-Textile waste composites) at the end of the first paragraph we have added a sentence explaining the effects of the presence of spray marks on the potential applications of composite materials. The added sentence related to this point can be easily identified by highlighting it in green in the revised manuscript.

  1. Line 408: Have you explored methods to minimize or eliminate the brown splay marks caused by liny gas bubbles during molding?

ANSWER

We thank the reviewer for this comment. As an answer to the question, we can say that it is really very interesting what the reviewer comments, and for future work we will keep it in mind, however in this work we have not considered any method to minimize or eliminate the spray marks caused by liny gas bubbles during molding.

  1. Line 411: How does the visual appearance of the composites change with varying amounts of textile waste and the use of silanes?

ANSWER

We thank the reviewer for this comment. As an answer to the question, we can say that as the percentage of textile waste increases, the composites develop darker brown shades, which are more pronounced with higher textile waste content. In the text we believe it is adequately explained.

  1. Line 414: How do the mechanical properties compare between composites with different visual appearances? Is there a correlation between visual defects and mechanical performance?

ANSWER

We thank the reviewer for this comment. As an answer to the question, we can say that we have found no correlation between visual defects and mechanical properties at work. However, this is an interesting study for future publications.

  1. Line 417: Could the improved compatibility and adhesion between the polyamide matrix and textile waste fibers be further quantified or visualized?

ANSWER

We thank the reviewer for this comment. As an answer to the question, we can say that improved compatibility and adhesion between the polyamide matrix and textile waste has been quantified through the characterization of the mechanical properties and visualized by means of using scanning electron microscopy (SEM) to observe the interfacial bonding at the microscopic level.

  1. Line 420: Are there any specific applications where the enhanced tensile strength of composites with up to 60 wt. % textile waste is particularly advantageous?

ANSWER

We thank the reviewer for this comment. As an answer to the question, we can say that the enhanced tensile strength of composites with up to 60 wt. % textile waste is particularly advantageous for applications requiring materials with both high strength and sustainability. In the construction sector, this composite can also be used in applications such as decking, cladding, and structural panels. Additionally, consumer goods, including furniture and sports equipment, can benefit from these composites.

  1. Line 423: How does the reduction in degradation onset temperature due to cotton presence affect the thermal stability and potential high-temperature applications of the composites?

ANSWER

We thank the reviewer for this comment. As an answer to the question, we can say that the reduction in degradation onset temperature due to the presence of cotton negatively affects the thermal stability and potential high-temperature applications of the composites. This limits the use of these composites in applications that require prolonged exposure to high temperatures, such as certain automotive parts and industrial components. Therefore, while the composites benefit from enhanced mechanical properties and sustainability, their thermal stability must be carefully considered when selecting them for high-temperature environments.

  1. Line 426: How does the incorporation of textile waste influence other surface properties such as abrasion resistance and surface hardness?

ANSWER

We thank the reviewer for this comment. As an answer to the question, we can say that the incorporation of textile waste into the polyamide matrix influences surface properties such as abrasion resistance and surface hardness. The paper reports that composites with a higher content of textile residues usually show an increase in surface hardness and, consequently, in abrasion resistance.

  1. Line 429: Have you conducted tests to measure the impact of increased hydrophilicity on the composites' resistance to other environmental factors like UV exposure or chemical exposure?

ANSWER

We thank the reviewer for this comment. As an answer to the question, we can say that it is really very interesting what the reviewer comments, and for future work we will keep it in mind, however in this work we have not considered any study of resistance to other environmental factors such as resistance to UV radiation or chemical agents.

  1. Line 432: What are the potential environmental benefits and challenges of using these composites in real-world applications, considering their hydrophilic nature and visual appearance?

ANSWER

We thank the reviewer for this comment. As an answer to the question, we can say that using composites of polyamide with textile waste fibers offers significant environmental benefits, including waste reduction, resource conservation, and a lower carbon footprint. However, challenges such as the hydrophilic nature of textile fibers, variability in fiber quality, recycling complexity, and potentially less uniform visual appearance need to be addressed. Strategies like surface treatments, fiber blending, advanced manufacturing techniques, and stringent quality control can help mitigate these challenges, making these composites a promising sustainable material option in real-world applications.

Round 2

Reviewer 1 Report

Comments and Suggestions for Authors

The paper is accepted in its current form.

Reviewer 2 Report

Comments and Suggestions for Authors

The revised manuscript can be accepted